# Cholesterol Metabolism: A Potential Therapeutic Target in Glioblastoma

**DOI:** 10.3390/cancers11020146

**Published:** 2019-01-26

**Authors:** Fahim Ahmad, Qian Sun, Deven Patel, Jayne M. Stommel

**Affiliations:** National Institutes of Health, National Cancer Institute, Radiation Oncology Branch, Bethesda, MD 20892, USA; fahim.ahmad@nih.gov (F.A.); qiansun1128@gmail.com (Q.S.); devenpatel7a@gmail.com (D.P.)

**Keywords:** glioblastoma, cholesterol, liver X receptor (LXR), brain, liver, metabolism, blood–brain barrier, low-density lipoprotein receptor (LDLR), sterol regulatory element binding protein (SREBP)

## Abstract

Glioblastoma is a highly lethal adult brain tumor with no effective treatments. In this review, we discuss the potential to target cholesterol metabolism as a new strategy for treating glioblastomas. Twenty percent of cholesterol in the body is in the brain, yet the brain is unique among organs in that it has no access to dietary cholesterol and must synthesize it de novo. This suggests that therapies targeting cholesterol synthesis in brain tumors might render their effects without compromising cell viability in other organs. We will describe cholesterol synthesis and homeostatic feedback pathways in normal brain and brain tumors, as well as various strategies for targeting these pathways for therapeutic intervention.

## 1. Pathological and Genetic Features of Glioblastoma

Glioblastoma (also called GBM) is the most common malignant primary brain tumor originating from glial cells [1]. Of the three types of gliomas (ependymomas, oligodendrogliomas, and astrocytomas), glioblastomas are WHO grade IV astrocytomas and have the worst prognosis. GBMs cause 225,000 deaths per year worldwide, and are diagnosed at an average age of 64 [2]. The median survival rate for GBM is 15 months from initial diagnosis even with the most current standard-of-care therapy, which consists of maximal surgical resection, radiation therapy, and adjuvant chemotherapy with temozolomide [3]. There are several obstacles to the development of efficient treatments against glioblastoma. Surgical resection of GBM is virtually impossible as these tumors are highly invasive and penetrate the normal brain. Full resection would require very fine tuned and precise imaging tools, which would enable the removal of invading tumor cells. GBM is highly resistant to cytotoxic drug regimens, including temozolomide, which only improves overall survival 2.5 months beyond radiation and surgery alone [4]. Therefore, new strategies are necessary to develop treatments that elicit durable responses in GBM patients.

Various rodent models have been developed for the accurate representation of preclinical GBM models, but those systems have multiple drawbacks such as immune system deficiency and incompatible stroma and microenvironment that might interfere with the testing new drugs [5]. Existing animal models fall under three categories. The first involves genetically engineered mouse models: for example, mice expressing v-src kinase driven by an astrocyte-specific Glial Fibrillary Acidic Protein (*GFAP*) promoter or *tp53*-null mice with astrocyte-specific loss of *NF1* both develop high-grade astrocytomas [6,7]. Mouse models of glioblastoma have also been generated using viruses expressing oncogenes injected into the mouse brain. For example, Pax3-Tv-a; Trp53 fl/fl mice injected with RCAS-PDGFB and RCAS-Cre virus, with or without RCAS-H3.3K27M, develop a tumor similar to diffuse pontine glioma [8]. A more recent technical development is the injection of patient-derived glioblastoma stem-like cells in immunocompromised mice. While many laboratories have adopted this technique for studying glioblastoma in vivo, two recent examples include injecting cells derived from isocitrate dehydrogenase 1 (*IDH1*)-mutant tumors into SCID (Severe Combined Immunodeficiency) mice to study vulnerability of this tumor genotype to 2-hydroxyglutarate depletion [9], and injecting cells from recurrent glioblastoma into NOD-SCID (Non-Obese Diabetic SCID) and NOG (NOD/Shi-scid/IL-2Rγnull) mice then treating them with a STAT3 (Signal Transducer and Activator of Transcription 3) inhibitor [10]. These animal models have led to many novel discoveries in glioma cell biology and metabolism, but thus far have not led to any new clinical advances.

Our most promising avenues for developing robust strategies to target GBM are likely to lie within recent efforts to genomically and proteomically catalog this disease. Indeed, a study published by The Cancer Genome Atlas (TCGA) Research Network in 2014 showed that the most frequently altered pathways in GBM are the RTK/PI3K/MAPK (90% of tumors), p53 (86%), and Rb pathways (79%) [11]. Because nearly all glioblastomas have at least one genomic alteration in the RTK/PI3K/MAPK axis, and because there are numerous small molecule inhibitors targeting this pathway that are already FDA-approved or at least in early-stage clinical trials, it appears inhibition of RTKs, PI3Ks, or MAPKs should be highly effective. Unfortunately, no RTK, PI3K, or MAPK pathway inhibitor thus far has improved patient survival beyond that of the current standard-of-care [4,12,13]. That said, the TCGA datasets have provided a trove of mutations, gene expression and proteomic profiles, and copy number variations that can be explored for novel therapeutic targets. A recent review published by Alphandery et al. has covered all the most recent glioblastoma treatments that are commercialized or under industrial development for glioblastoma treatment [2].

## 2. Dysregulated Metabolic Pathways in Glioblastoma

One strategy lacking from conventional GBM therapies is metabolic pathway targeting. Significant pre-clinical research has been done in this field to understand how tumor cells differ from their counterparts. For example, work from Calvert et al. using in silico and wet-bench analyses shows that non-mutated isocitrate dehydrogenase 1 (IDH1) is commonly overexpressed in the primary GBM and genetic and pharmacological inactivation leads to attenuation of GBM tumor growth [14]. Large-scale genomics studies have also revealed that IDH1 mutations occur in gliomas even before other mutations such as TP53 [15,16]. Glutamate and alpha-ketoglutarate are fundamental metabolites that are necessary for growth and proliferation of GBM cells. A study published by Franceschi et al., showed that glutamate dehydrogenase 2 (GLUD2, an enzyme responsible for glutamate oxidative deamination), inhibits GBM cell growth and could be a target to control tumor progression [17]. In contrast with previous work showing the importance of the tricarboxylic acid (TCA) cycle and amino acid metabolism in GBM, in this review we will discuss the role of cholesterol metabolism in influencing tumor cell growth and proliferation and as a potential therapeutic target.

## 3. Feedback Pathways in Cholesterol Biosynthesis

Cholesterol is critical for cell growth and function. It is important not only as a component of the plasma membrane and lipid rafts, it also serves as a precursor for steroid hormones, bile acids, and vitamin D [18,19,20]. In the last few years, researchers have explored the impact of cholesterol metabolism on immune response. For example, Yang et al. demonstrated that by modulating cholesterol metabolism in CD8 + T cells, higher antitumor activity could be achieved [21]. Another study published by Wang et al. showed that an analog of cholesterol can act as a negative regulator for TCR signaling [22]. Cells usually obtain cholesterol via different mechanisms. One is through direct synthesis via the transcriptional activity of sterol regulatory element binding proteins (SREBPs), which promote the transcription of enzymes involved in cholesterol and fatty acid biosynthesis pathways, including HMG-CoA reductase (HMGCR) [23]. Ground-breaking work by Brown and Goldstein, who later won the Nobel prize for the discovery, defined cholesterol as a key molecule regulating its own synthesis via the activation of a negative feedback mechanism (Figure 1) [24]. In the presence of cholesterol, SREBP is sequestered in the endoplasmic reticulum (ER) by the SREBP cleavage-activating protein (SCAP), whose function is inhibited by the ER-resident insulin-induced gene (*INSIG1*, *INSIG2*) [25]. When cholesterol levels drop below homeostatic levels, SCAP is separated from INSIGs via a conformational change [26] and carries SREBPs into the Golgi where their active sites are cleaved and activated as transcription factors [27]. SREBPs include three members: SREBP1a, SREBP1c, and SREBP2 [28]. SREBP2 is encoded by the *srebf2* gene and SREBP1a and SREBP1c are encoded by the *srebf1* gene. SREBP1c regulates the transcription of the genes that are associated with biosynthesis of fatty acids; SREBP2 mainly regulates genes involved with cholesterol biosynthesis. Activity of SREBP1a partially overlaps between SREBP1c and SREBP2.

Cholesterol can be enzymatically modified to form metabolites such as oxysterols. One species of oxysterol, 25-hydroxycholesterol (25-OHC), has been shown to be associated with suppression of proliferation [29]. The suppression in proliferation can be rescued by the addition of exogeneous cholesterol, indicating that the presence of 25-OHC inhibits SREBP activation and suppresses cholesterol biosynthesis [26]. 25-OHC also has anti-inflammatory effects: animal models have demonstrated that it inhibits transcription and inflammasome-mediated activation of interleukin-1β (IL1B) by inhibiting the activation of SREBPs [30]. Of interest for this review, recent studies have shown that 25-OHC also suppresses the immune response in the human glioblastoma cell line, U87-MG, and thus might increase tumor growth by modulating the immune system [31].

Oxysterols also regulate the activity of liver X receptors (LXRs), which are nuclear receptors that are activated by oxysterols. LXRα is expressed highly in liver, adrenal glands, intestine, adipose tissues, macrophages, lungs, and kidney, while LXRβ is ubiquitously expressed [32]. The LXRs maintain cholesterol homeostasis by maintaining the balance between biosynthesis, uptake via low density lipoprotein receptors (LDLRs), and efflux via the ATP-binding cassette transporters, ABCA1 and ABCG1 (Figure 1). LXRs inhibit LDLR protein expression through induction of an E3 ligase that ubiquitinates LDLRs called IDOL (inducible degrader of LDLR, which is encoded by the *MYLIP* gene [33]. The importance of LXRs in the central nervous system and in brain development was recently reviewed by Courtney et al. [34].

## 4. Cholesterol in the Normal Brain

The brain contains about 20% of the cholesterol of the whole body, rendering it the most cholesterol-rich organ [35]. Previous studies have shown the possibility of circulating cholesterol, in some manner, affecting the function of the central nervous system (CNS): for instance, low circulating cholesterol levels might be linked with violent behavior [36,37,38]. It is also postulated that brain development and intelligence is related to the levels of circulating cholesterol of a newborn infant [39,40]. However, a series of experiments conducted later provide no direct evidence for lipoprotein cholesterol crossing the blood–brain barrier (BBB) [41,42,43,44]. Thus, it is believed that the BBB prevents the entry of lipoproteins into the brain, and the accumulation of brain cholesterol is mainly achieved through de novo synthesis. In addition, several proteins related to cholesterol metabolism have been found in the brain, such as the apolipoproteins ApoE and ApoAI, LDLRs, scavenger receptor class B type I (SRB1, encoded by the *SCARB1* gene), and ABC transporters. Whether they play the same role in the brain as in other organs is still under investigation.

Cholesterol metabolism in the brain is well-regulated through the coordinating work of a series of proteins. The mechanisms of acquiring cholesterol include de novo synthesis and uptake of cholesterol from the external environment by LDLR, SRB1, and Niemann–Pick C1-like protein (NPC1L1) [45]. The synthesis of cholesterol in brain, as in other organs, starts from the conversion of acetyl-CoA to 3-hydroxy-3-methylglutaryl-CoA with HMG-CoA as the rate-limiting enzyme. SREBPs in the endoplasmic reticulum sense the levels of cholesterol and regulate the activity of HMG-CoA [46]. Meanwhile, the uptake of cholesterol can be achieved through taking up lipoproteins from the extracellular environment. One example is the binding of particles that contain ApoE to LDLR, which are then processed through the clathrin-coated pit pathway to endosomes and lysosomes [47]. Moreover, Niemann–Pick type C1 and C2 are also required to move cholesterol to the plasma membrane [48]. The excretion of cholesterol out of the cell may be driven by the chemical gradient between leaflet and lipoprotein receptors in the plasma membrane. Cholesterol can also be exported from the cells by ABC transporters. Hundreds of ABC transporters have been found in both prokaryotes and eukaryotes. Of the 48 ABC transporters in human genome, 13 ABC transporters (ABCA1, ABCA2, ABCA3, ABCA4, ABCA7, ABCA8, ABCB1, ABCB4, ABCD1, ABCD2, ABCG1, ABCG2, and ABCG4) have been studied in human brain [49]. As mentioned previously, LXRα and LXRβ can regulate the expression of ABCA1 and ABCG1 to control the efflux of cholesterol and phospholipids. It was found that LXR agonists enhance cholesterol efflux in astrocytes [50]. In addition, cholesterol in the brain and other organs can be hydroxylated by various enzymes to form hydroxylated sterol molecules and excreted from cells by diffusion [36]. Sterols in the brain, especially in the adult, are essentially non-esterified cholesterol. The presence of cholesteryl esters in the brain correlates with the occurrence of disease, such as multiple sclerosis [51].

## 5. Cholesterol Metabolism in Embryonic vs. Adult Brain

About 70–80% of the cholesterol in the brain is found in myelin sheaths, and the rest exists in the membranes of cell organelles. The total cholesterol levels in the brain are low during the perinatal period and at the time of birth, which is thought to limit the development of the CNS to allow the baby’s head to pass through the pelvis and birth canal. The cholesterol concentration of the whole brain increases from about 6 mg/g at birth to 23 mg/g in young adults [47]. Animal experiments demonstrate that the rate of cholesterol synthesis in the brain correlates with the rate of cholesterol accumulation and the final concentration of cholesterol found in those regions [44,52,53,54]. Therefore, de novo synthesis, not exogenous lipoprotein cholesterol import, is the major pathway for the accumulation of cholesterol in the brain during early development. Furthermore, studies also show that cholesterol synthesis during early development is accompanied by the synthesis of myelin basic protein and cerebroside [54]. These findings indicate that the accumulation of cholesterol in the brain associates with myelination. The synthesis of brain cholesterol slows down when neuron myelination is complete in adulthood, and the half-life of cholesterol lengthens to between 6 months and 5 years [55]. In contrast with early development when cholesterol is synthesized de novo, in adulthood nerve cell bodies can take up cholesterol using LDLR-related protein. Cell culture studies have shown that media from glial cells which contains cholesterol and ApoE stimulates the extension of axons, and LDLR inhibitors prevent the extension [56]. Furthermore, the formation of synapses of retinal ganglion cells requires glial cells to produce cholesterol and ApoE [57,58]. Therefore, cholesterol is probably mainly synthesized in both neurons and glial cells in the adult brain. 

## 6. Cholesterol Excretion from the Brain

To maintain cholesterol homeostasis in the brain, especially during adulthood, the excretion of cholesterol is important. 24(S)-hydroxycholesterol is the major hydroxylated sterol excreted from brain [59]. In addition, the enzyme that synthesizes this oxysterol, cholesterol 24-hydroxylase (encoded by the gene *CYP46A1*), is primarily expressed in brain compared to other organs. The hydroxylated cholesterol can cross the BBB and go to the liver to be converted to bile acids and excreted from the body. 

## 7. Cholesterol Metabolism in the Liver vs. Brain

The liver plays an important role in cholesterol metabolism. In contrast with brain, hepatic cholesterol can be obtained from the diet. Dietary cholesterol is obtained by intestinal epithelial cells via endocytosis. Cholesterol can then be esterified and loaded into nascent chylomicrons together with triacylglycerol. The chylomicrons are released from intestinal cells into the circulation by the lymphatics [60]. Triacylglycerol in the chylomicrons is hydrolyzed by lipoprotein lipase in blood vessels, and the cholesterol left behind in the chylomicron remnants are taken up and utilized by the liver [60,61]. The cholesterol from liver and dietary origins can be packed into particles of very low-density lipoproteins (VLDLs), which leave the liver and transport cholesterol to other tissues [62]. Another difference between liver and brain cholesterol metabolism is that cholesterol can be recycled through enterohepatic circulation, which does not exist in the brain (Figure 2). Cholesterol can be oxidized in the liver to form bile acids which along with cholesterol is excreted from the liver into the bile [63,64]. The excretion of bile acid involves ABCB11 [65]. Only about 5% of bile acids are lost in the feces, and the rest are reabsorbed into enterocytes. Bile acid is important for the digestion and absorption of dietary fats.

In summary, cholesterol is involved in cell membrane formation and signaling, and is the precursor of many steroid molecules such as steroid hormones, vitamins, and bile salts. Thus, the metabolism of cholesterol is tightly regulated throughout the body. Although the liver is the primary organ regulating cholesterol homeostasis, the brain cannot uptake cholesterol from peripheral blood and diet due to the BBB (Figure 2). Brain cholesterol is primarily derived from de novo synthesis, and cholesterol levels start to accumulate after birth. Upon reaching adulthood, brain cholesterol levels are maintained at constant levels. Therefore, the excretion of cholesterol from the brain becomes more active in adulthood. Brain cholesterol can be hydroxylated and pass through the BBB to form bile acids in the liver. Moreover, for some types of nerve cells cholesterol must be acquired through the binding of low-density lipoproteins LDLs and LDLRs since cholesterol is mainly synthesized in glial cells and neurons in the adult brain. Disturbed homeostasis of brain cholesterol can lead to diseases such as dementia.

## 8. Cholesterol Metabolism Pathways Are Altered in Brain Tumors

The brain has different ways to satisfy the requirements of cholesterol compared to peripheral organs. An epidemiological study investigated the relationship between dietary intake of cholesterol and the incidence of cancer, and found that high dietary cholesterol levels increase the risk of several cancers including stomach, colon, rectum, pancreas, lung, breast, kidney, bladder, and non-Hodgkin’s lymphomas, but not brain tumors [66]. This result is not surprising: since cholesterol cannot pass the BBB, high plasma cholesterol levels are unlikely to affect cholesterol metabolism in the brain [66].

The brain obtains cholesterol primarily through de novo synthesis, which involves the mevalonate and Bloch and Kandutsch-Russell pathways [67,68,69]. Taking advantage of the Glioblastoma Bio Discovery Portal [70], our group found a correlation between upregulation of mevalonate and cholesterol pathway and poor survival of GBM patients [71]. Mechanistic studies demonstrated that densely-plated glioma cells increase the synthesis of cholesterol by enhancing oxygen consumption, glycolysis and the pentose phosphate pathway, and pharmacological inhibitors acting downstream of the mevalonate pathway induce glioma cell death [71]. Moreover, the study from our group also found that densely plated normal astrocytes but not tumor sphere glioma cells downregulate genes in the cholesterol biosynthetic pathway including farnesyl diphosphate synthase, farnesyl-diphosphate farnesyltransferase 1, and squalene epoxidase, (*FDPS*, *FDFT1*, and *SQLE*, Figure 3) [71]. A study conducted later by Kim et al. showed that inhibition of FDPS by pharmacological inhibitors and siRNA (small interfering RNA) prevents the formation of secondary spheres of glioma stem cells, and *FDPS* mRNA was associated with malignancy in glioblastoma patients [72]. In addition to mRNA and protein levels, Abate et al. demonstrate that the activity of FDPS is also upregulated in GBM tumor tissue [73].

Another study demonstrates that GBM is dependent proper cholesterol homeostasis for survival: instead of inhibiting the synthesis of cholesterol, Villa et al. tested the effect of LXRs on treating GBM [74]. LXR is a transcription factor that facilitates the efflux of cholesterol by increasing ABCA1 expression. Villa et al. showed that limiting cholesterol levels by treating cells with LXR agonists induced glioma cell death. In vivo experiments showed that LXR agonists inhibited GBM growth and prolonged the survival of mice [74]. As a key transcription factor in the regulation of sterol homeostasis, other groups have evaluated the effects of SREBPs on GBM development [75,76]. Lewis et al. showed that under hypoxia and serum-deprivation conditions, SREBP is upregulated to maintain the expression of fatty acid and cholesterol biosynthetic genes in GBM cells, and inhibiting SREBP activity under hypoxia led to GBM cell death [75]. These studies demonstrate that cholesterol metabolism pathways are upregulated in GBM patients and targeting cholesterol metabolism and/or homeostasis may be a promising strategy in treating GBM.

## 9. Targeting Cholesterol Metabolism as a Glioblastoma Therapy

Cancer cells have an increased demand for cholesterol and cholesterol precursors. Therefore, a reasonable assumption is that prevention of tumor-cell growth can be achieved by restricting either cholesterol availability or cholesterol synthesis [77]. Loss of cholesterol feedback inhibition mechanisms that regulate cholesterol synthesis is an important feature of malignant transformation. The cholesterol synthesis pathway has numerous proteins that are potential targets to disrupt cancer progression [78]. The therapeutic potential of targeting these cholesterol synthesis genes is under preclinical investigation [79,80]. The unique metabolic requirements of the brain might make glioblastoma particularly suitable for cholesterol pathway targeting [71].

Liver X-receptors (LXRs) act as transcriptional master switches that coordinate the regulation of genes such as *ABCA1* and *ABCG1*, which are involved in cellular cholesterol homeostasis [81,82] LXR-623 is a synthetic ligand for LXRα and β that upregulates *ABCA1* and *ABCG1* expression in blood cells [83]. In a study published by the Mischel lab, LXR-623 killed GBM cell lines in an LXR β- and cholesterol-dependent fashion but not healthy brain cells. Upon further investigation of LXR-623, the group found that the drug penetrated the blood–brain barrier and retained its anticancer activity. In addition, mice harboring GBM tumors derived from human patients and treated with LXR-623 had substantially reduced tumor size and improved survival [74].

Statins are HMG-CoA reductase inhibitors that have anti-tumor effects and synergize with certain chemotherapeutic agents to decrease tumor development [80,84]. Several clinical trials have examined the potential chemo-preventive and therapeutic efficacy of different formulations of statins, including simvastatin, pitavastatin, and lovastatin [85,86,87,88,89]. For example, a short-term biomarker study showed lowered breast cancer recurrence in simvastatin-treated patients through the reduction of serum estrone sulfate levels [90]. Moreover, pitavastatin reduces GBM tumor burden in xenografts [86]. Bisphosphonates and tocotrienols are another class of drugs which act as downstream inhibitors of the cholesterol synthesis pathway. Early investigations have shown that they can slow down cancer cell and tumor growth similar to statins [91].

Protein geranylgeranylation, a branch of the cholesterol synthesis pathway, was also found to be essential for maintaining stemness of basal breast cancer cells and to promote human glioma cell growth. GGTI-288, an inhibitor of the geranylgeranyl transferase I (GGTI) reduced the cancer stem cell subpopulation in primary breast cancer xenografts [79,92]. Moreover, high dependency of malignant glioma cells on the isoprenoid pathway for post translational modification of intracellular signaling molecules make this pathway a potential candidate for drug targeting. Recent work published by Ciaglia et al. shows antitumor activity of N6-isopentenyladenosine (iPA) on glioma cells. Its mechanism of action is primarily driven through AMPK-dependent epidermal growth factor receptor (EGFR) degradation, which further adds value to its candidacy as a potent antitumor drug [93,94]. Thus, multiple preclinical studies demonstrate that targeting the cholesterol synthesis pathway could be useful for modulating cancer growth, either through directly inhibiting cholesterol synthesis, or though inhibiting the production of mevalonate pathway-derived moieties used in protein post-translational modifications of oncogenes.

## 10. Cholesterol and its Derivatives as Anti-Glioblastoma Agents

Steroids and their derivatives or triterpene precursors such as betulinic, oleanolic, and ursolic acids and stigmasterol have shown strong anti-cancerous properties [95,96,97]. Cholesterol derivatives, for example, sodium cholesteryl sulfate, cholesteryl chloride, cholesteryl bromide, and cholesteryl hemi-succinate, have been reported to possess inhibitory activities against DNA polymerase and DNA topoisomerase and to inhibit human cancer cell growth [98].

Another promising approach in use is a cholesterol-based anticancer agent containing carborane as the anticancer unit for boron neutron capture therapy (BNCT). The cholesteryl 1,12-dicarba-*closo*-dodecaborane 1-carboxylate (BCH) mimics the native cholesteryl ester in structure and was found to be effectively taken up by brain glioma cells in vitro. BNCT delivers boron-10 packed in liposomes. To increase the delivery with an enhanced specificity to tumor tissue, these boron-10 were packed inside cholesterol-anchored folate in EGFR-folate receptor targeted liposomes or consist of cholesterol-anchored anti EGFR antibodies. Once the boron-10 is delivered to its designated location, low energy radiation is passed in the form of a thermal neutron that triggers fission reactions, resulting in the production of high linear energy transfer (LET) α-particles which are highly lethal to the cells [99].

## 11. Cholesterol-Based Intracerebral Delivery of Chemotherapeutics in Brain Tumors

The absence of compelling treatment alternatives results in poor prognosis of glioblastoma. It is vital to deliver adequate amounts of therapeutic agents to the brain tumor site. However, delivery of therapeutic agents to the tumor site is technically very challenging due to the presence of the BBB [100]. Many attempts have been made to overcome this. It is well established that LDL receptors are present on the BBB capillary endothelial cells and, therefore, could be utilized for transporting cholesteryl-based or other compounds to the brain.

Conjugation of a cholesterol moiety to an active medicinal compound for either cancer diagnosis or treatment is an attractive approach for targeted drug delivery. Approaches to intracerebrally administer agents within the brain parenchyma through local delivery to tumor tissue are on the rise [101]. The advantage of this approach is it results in high drug concentrations at the tumor site with restricted exposure to non-neoplastic tissues and organs.

## 12. Conclusions

The unique metabolic demands and dysregulated metabolism of GBM makes it particularly suitable for cholesterol pathway targeting [71,97,102]. Genomic analyses performed by the TCGA provide correlative evidence suggesting an involvement of the cholesterol homeostasis pathways in cancer development, especially in glioblastoma [103]. A vast number of genetic and phenotypic alterations in cholesterol homeostasis pathways have been identified in cancer cells [104]. These include increases in gene copy numbers and upregulation of cholesterol synthesis gene expression, enhanced cholesterol import by LDL receptors, and decreased transport of cholesterol, all of which promote increased cellular cholesterol levels to aid cancer cell proliferation [104,105,106,107].

First and the foremost, the genetic alterations influencing the cholesterol pathways in cancer development need further investigation. Many cholesterol synthesis genes or mitochondrial cholesterol importers are upregulated, however their effects on cancer development remain unknown. Out of these, several genetic alterations were associated with known chromosomal amplification sites that harbor well-characterized oncogenes. For example, *HMGCS2* and *NOTCH2,* and *SQLE* and *MYC*, were co-localized to the same amplicons. Possibly, oncogenes and cholesterol synthesis genes act in tandem to promote disease progression. However, limited success has been achieved in restricting tumor growth by targeting these critical genes with their pharmacological inhibitors (Table 1). Due to a lack of evidence to support their efficacy in treating different forms of tumors, this needs further exploration. A second question needing attention is whether tumors could be classified into subclasses based on genetic abnormalities occurring in cholesterol homeostasis genes. This might facilitate development of precision medicine-based approaches for treating subgroups of cancer. For examples, the efficacy of statins, squalene synthesis inhibitors, farnesyl or geranylgeranyl transferase inhibitors might be more effective for certain patients with characteristic genetic signatures.

Conjugation of the cholesterol moiety to an active medicinal compound for cancer disease diagnosis and treatment is an attractive approach for targeted drug delivery. Several anticancer agents have also shown promise in cholesterol formulation for brain delivery. However, a greater understanding of the biodistribution and pharmacokinetics of these cholesteryl drug conjugates is essential for their practical use. For example, once active drug molecules are conjugated to cholesterol moieties, their chemical properties such as hydrophilicity/lipophilicity and molecular weight are significantly altered, which can change their biodistribution, pharmacokinetics, and efficacy. Hence, a thorough analysis of the interactions of these cholesteryl conjugates with cells, receptors, and membranes is necessary prior to their use in a clinical setting. Nevertheless, the use of cholesteryl drug conjugates for targeted delivery provides a novel approach for treating a variety of cancers, including glioblastoma.

In summary, it appears that deregulation of cholesterol homeostasis is an important contributing factor to cancer development and particularly to brain tumors. Studies are needed to link population-based epidemiological data, results from the TCGA database, and preclinical mechanistic evidence to more thoroughly dissect the involvement of cholesterol in cancer development, which would be helpful in devising new strategy for therapeutic intervention.

## Figures and Tables

**Figure 1 cancers-11-00146-f001:**
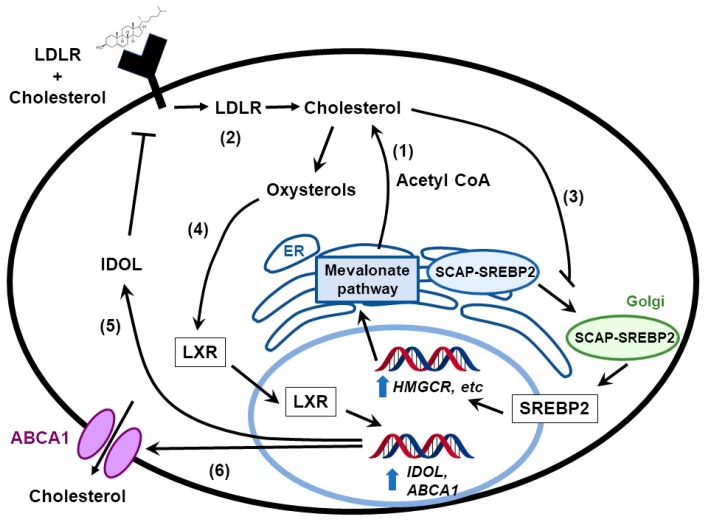
Cholesterol homeostasis in normal cells. Cells obtain cholesterol primarily through one of two mechanisms: (1) by synthesizing it de novo from acetyl CoA generated from glycolysis and (2) through exogenous uptake by low density lipoprotein receptors (LDLR). Cholesterol can negatively regulate its own levels through (3) the inhibition of proteolytic processing and nuclear import of sterol regulatory element binding proteins (SREBP2), leading to a decrease in activity in the mevalonate pathway or (4) through its conversion to oxysterols that activate liver X receptors (LXRs). LXRs lower cellular cholesterol levels by (5) inducing the transcription of the E3 ubiquitin ligase, *IDOL*, which ubiquitinates LDLR, and (6) by upregulating expression of the cholesterol efflux pump, *ABCA1*. SCAP: SREBP cleavage-activating protein; ER: endoplasmic reticulum.

**Figure 2 cancers-11-00146-f002:**
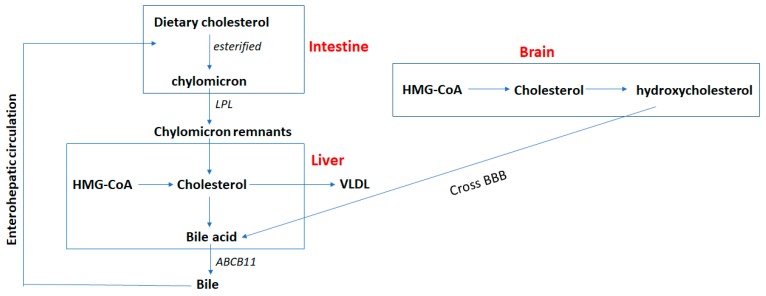
Cholesterol metabolism in liver vs. brain. The brain obtains cholesterol exclusively from de novo synthesis. On the contrary, hepatic cholesterol can be obtained by de novo synthesis and through dietary intake. Dietary cholesterol can be esterified and loaded into chylomicrons in the intestine. The chylomicrons are released into circulation and hydrolyzed by lipoprotein lipase (LPL) to form chylomicron remnants. Cholesterol left behind in the chylomicron remnants are taken up and utilized in the liver. The cholesterol synthesized in liver and from dietary origins can be packed into very low-density lipoprotein (VLDL) and exported from liver. Cholesterol can also be oxidized in the liver to form bile acids which excreted from liver into the bile via the ABCB11 transporter. Cholesterol in the brain can be hydrolyzed to form hydroxycholesterol which crosses the blood–brain barrier (BBB) and goes to the liver to be converted to bile acid. Cholesterol in the liver can be recycled through enterohepatic circulation, which does not exist in the brain. About 5% of the bile acids are lost in the feces, and the rest are reabsorbed into enterocytes.

**Figure 3 cancers-11-00146-f003:**
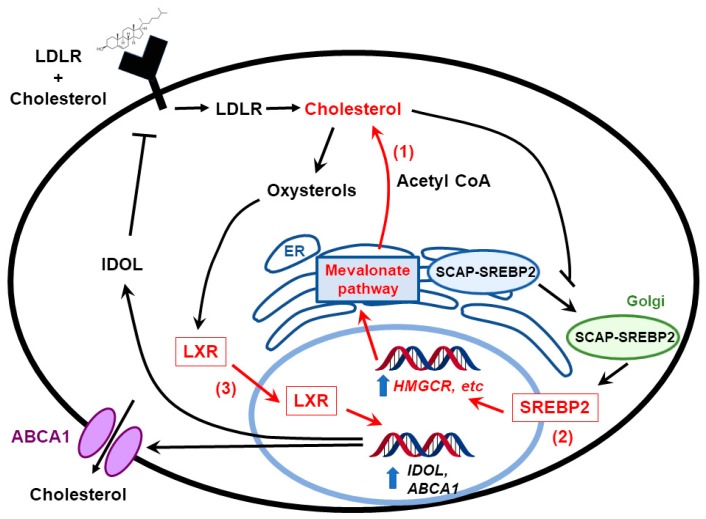
Cholesterol homeostasis in glioblastoma cells. Glioblastoma cells maintain cholesterol under conditions in which normal cells turn it off through multiple mechanisms of dysregulation (highlighted in red). They keep cholesterol biosynthesis on by constitutive activation of the mevalonate pathway (1), and by upregulating SREBPs under hypoxia (2). They are also highly dependent on appropriate levels of LXR activity—hyperactivating LXR with synthetic agonists overstimulates ABCA1 expression and cholesterol efflux, killing glioblastoma (GBM cells) (3). In sum, this provides them with cholesterol in an organ that is blocked from obtaining it from the circulation due to the blood-brain barrier.

**Table 1 cancers-11-00146-t001:** Commercial drugs targeting cholesterol pathways.

Drug	Mechanism
Ciprofibrate	PPARα agonist
Clofibrates	PPARα agonist
Fenofibrate	PPARα agonist
Gemifibrozil	PPARα agonist
Anacetrapib	CETP inhibitor
Avasimibe	ACAT inhibitor
Berberine	Increases LDLR expression
Lapaquistat acetate	FDFT1 inhibitor
Ezetimibe	NPC1L1 inhibitor
Atorvastatin	HMGCR inhibitor
Fluvastatin	HMGCR inhibitor
Pitavastatin	HMGCR inhibitor
Rosuvastatin	HMGCR inhibitor
Simvastatin	HMGCR inhibitor
Pitavastatin	HMGCR inhibitor

PPARα = peroxisome proliferator activated receptor alpha; CETP = cholesteryl ester transfer protein; ACAT = sterol O-acyltransferase 1; LDLR = low density lipoprotein receptor; FDFT1 farnesyl-diphosphate farnesyltransferase 1; NPC1L1 = NPC1 like intracellular cholesterol transporter 1; HMGCR = 3-hydroxy-3-methylglutaryl-CoA reductase.

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
