# Peer review of "Cholesterol Metabolism: A Potential Therapeutic Target in Glioblastoma"

_cancers, 2019, doi:10.3390/cancers11020146_

Round 1

Reviewer 1 Report

The authors reviewed an actual and interesting chapter of cancer metabolism. Concerning GBM, this is of great interest as it is the most fatal cancer among human malignancies and so new therapeutic strategies are needed. 

 While Fahim Ahmad et al. offer a clear, nice and complete description of the cholesterol biosynthesis in physiological setting, some interesting aspects in pathological context should be integrated in order to corroborate their statements.

For example:

a) authors said that RTKs, PI3Ks, or MAPKs should be highly effective target in GBM progression, please clarify the importance for example  of  the isoprenoid moieties for correct localization and signaling of EGFR and possibly new strategies to counteract it (e.g antiglioma effect of isoprenoid derivative through EGFR downregulation etc...Ciuaglia et al., Int J Cancer. 2017; Br J Pharmacol. 2017 Jul;)

b) the authors also report a correlation between upregulation of mevalonate and cholesterol pathway and poor survival of GBM patients. Among these, FDPS mRNA was associated with malignancy in glioblastoma patients[65]. Interestingly, in GBM it has been recently described not only  a Deregulation of expression but also of enzyme activity of Farnesyl Diphosphate Synthase (FDPS) in Glioblastoma.(Abate et al., Sci Rep. 2018). 

Through these kind suggestions, I feel the manuscript which already sounds very well, might be suitable for final publication.

Best regards

Author Response

Reviewer 1:

a)      authors said that RTKs, PI3Ks, or MAPKs should be highly effective target in GBM progression, please clarify the importance for example  of  the isoprenoid moieties for correct localization and signaling of EGFR and possibly new strategies to counteract it (e.g antiglioma effect of isoprenoid derivative through EGFR downregulation etc...Ciuaglia et al., Int J Cancer. 2017; Br J Pharmacol. 2017 Jul;)

 ·         Thank you for suggesting these references.  We have added a mention of this work in section 9.

b)      the authors also report a correlation between upregulation of mevalonate and cholesterol pathway and poor survival of GBM patients. Among these, FDPS mRNA was associated with malignancy in glioblastoma patients[65]. Interestingly, in GBM it has been recently described not only  a Deregulation of expression but also of enzyme activity of Farnesyl Diphosphate Synthase (FDPS) in Glioblastoma.(Abate et al., Sci Rep. 2018).

 ·         We thank the reviewer for suggesting the addition of this reference.  It is now included in section 8.

Reviewer 2 Report

This review paper described about the current knowledge regarding the glioblastoma (GBM) therapy targeting or utilizing intracellular cholesterol metabolism of GBM. The authors first described about the differences of cholesterol metabolic feature between normal tissue and tumors including GBMs and explain why cholesterol metabolism is an attractive and potent therapeutic target in GBM therapy. In addition, this paper also reviewed in detail about the recent clinical trials for GBMs utilizing either the cholesterol metabolism-targeting drugs as antitumor agents or the drug delivery systems applying GBM-specific cholesterol metabolism. On the whole, this review paper is well-written and condensed.

# Comments:

1)      This paper carefully mentioned about cholesterol metabolism of normal tissue in first half of the text, however, considering the theme of this paper, it would be better to shift the contents to describe (emphasize) further about the cholesterol metabolism of GBM cells.

2)     For the same reasons, the subtitles of this article should emphasize about “glioblastomas” not “Cancers” or “Tumors”.

3)     Demonstrating the schematic illustration of the distinctive cholesterol metabolism of GBM cells by the figure is also important.

4)     It is interesting and important to know the therapeutic roles of cholesterol metabolism in recurrent GBMs after standard therapy (surgery followed by irradiation and temozolomide treatment), because these kinds of novel therapy against malignancies are usually first applied to the relapsed cases after first line standard therapy.

Author Response

Reviewer 2:

1)      This paper carefully mentioned about cholesterol metabolism of normal tissue in first half of the text, however, considering the theme of this paper, it would be better to shift the contents to describe (emphasize) further about the cholesterol metabolism of GBM cells. 

 ·         We agree that in a therapeutic context, the aberrations of cholesterol metabolism in GBM tumors is highly interesting.  However, because this review is in an issue devoted to GBM biology and not cholesterol metabolism, we were concerned that many of the readers might not have much background knowledge of cholesterol metabolic pathways in normal cells and therefore included this for comparison to GBM.

2)      For the same reasons, the subtitles of this article should emphasize about “glioblastomas” not “Cancers” or “Tumors”.

 ·         We altered the section headings of this review from mentions of “cancer” to “glioblastoma” at this reviewer’s request.

3)      Demonstrating the schematic illustration of the distinctive cholesterol metabolism of GBM cells by the figure is also important.

 ·         We thank you for this suggestion.  We have clarified cholesterol homeostasis in normal cells in Figure 1, and have now included a new Figure 3 which contrasts its points of dysregulation in GBM.

4)      It is interesting and important to know the therapeutic roles of cholesterol metabolism in recurrent GBMs after standard therapy (surgery followed by irradiation and temozolomide treatment), because these kinds of novel therapy against malignancies are usually first applied to the relapsed cases after first line standard therapy.

 ·         We are not aware of any studies examining cholesterol metabolism in recurrent GBMs after standard therapy, though we agree with this reviewer that this might be a very interesting novel treatment after GBM relapse. 

Reviewer 3 Report

The work of Ahmad and co-authors clearly describes the metabolic changes occurring during neoplastic transformation. It is well written and it revises the present literature. Although some criticism can be raised:

1-page 1, lane 31: The authors indicate that “various rodent model” has been developed to a representation of GBM. I do believe that at least one or two examples are needed.

2-page 3, lane 93: Describing the role of 25-OHC, authors do not mention the role of this metabolite in immunomodulation. Please describe how 25-OHC modulate IL-1B production and its role in inflammation, see Simon 2014 and Tricarico et al., 2017.

3-page 5 paragraph “Cholesterol metabolism in the liver vs. brain” is very interesting, but I do think that a figure that describes and summarizes the mechanisms described there can be very helpful for the reader

4-page 7 lanes 267: “Boron Neutron capture therapy”. Please briefly describe this technique. 

Author Response

1-page 1, lane 31: The authors indicate that “various rodent model” has been developed to a representation of GBM. I do believe that at least one or two examples are needed. 

 ·         We have now clarified this statement and included discussion of multiple GBM animal models in Section 1.

2-page 3, lane 93: Describing the role of 25-OHC, authors do not mention the role of this metabolite in immunomodulation. Please describe how 25-OHC modulate IL-1B production and its role in inflammation, see Simon 2014 and Tricarico et al., 2017.

·         We have added mention of the findings of these two references in Section 3.

3-page 5 paragraph “Cholesterol metabolism in the liver vs. brain” is very interesting, but I do think that a figure that describes and summarizes the mechanisms described there can be very helpful for the reader.  

    ·         We thank the reviewer for this suggestion.  We now include Figure 2, with briefly summarizes and contrasts cholesterol metabolism in normal liver versus normal brain.

4-page 7 lanes 267: “Boron Neutron capture therapy”. Please briefly describe this technique.

 ·         We have described Boron Neutron capture therapy in more detail in section 10.